# Evidence of allocentric spatial learning in male rats with large lesions of the hippocampus

**Jordan A. Webb**[1]*, **Sebastien Paquette**[1], **Neil M. Fournier**[1], **Michael G. Reynolds**[1], **Liana E. Brown**[1,2], **Hugo Lehmann**[1]

**1** Psychology Department, Trent University, Peterborough, Ontario, Canada, **2** Department of Kinesiology, Trent University, Peterborough, Ontario, Canada

* jordanwebb@trentu.ca

## Abstract

The hippocampus (HPC) is the neural substrate of viewpoint-invariant cognitive maps, also known as allocentric spatial representations. Lesions of the HPC disrupt performance on allocentric tasks like the Morris Water Task (MWT), in which rodents must learn and recall the location of a platform submerged within a circular pool. Success in finding the hidden platform from any start point requires integrating multiple types of information, such as discerning its location relative to fixed (allocentric) environmental cues. Rats with HPC lesions, however, may show improvement in the MWT over repeated swim trials by resorting to alternative search strategies based on body-centered (egocentric) cues. Here, we investigated whether HPC lesion size correlates with allocentric impairments in the MWT. Using swim path classification alongside standard performance measures, we analyzed an archival dataset of 53 HPC lesion and 15 control rats trained under the same protocol. All rats showed evidence of learning, but the HPC group demonstrated impairment relative to the control group. Further analysis revealed that control rats shifted to a persistent allocentric search strategy by the fifth trial. The HPC rats shifted to persistent strategy use by the sixth trial, but not necessarily to an egocentric strategy. Interestingly, a subset of the HPC rats developed and maintained an allocentric strategy. Performance was not correlated with lesion size. These findings suggest that a subgroup of rats with HPC lesions, even nearly complete ones, can learn and remember allocentric spatial information in the MWT. This highlights the potential role of other brain regions in supporting spatial learning and memory in the absence of the HPC.

## Introduction

The hippocampus (HPC) is a structure within the medial temporal lobe that plays a crucial role in spatial memory [1]. It supports the formation of allocentric spatial representations, meaning viewpoint-independent cognitive maps in which locations are encoded relative to other environmental landmarks [2,3]. The Morris Water Task

**Data availability statement:** All relevant data are within the manuscript and its Supporting Information files.

**Funding:** This research was supported by a Discovery Grant (RGPIN-2018-04901) from the Natural Sciences and Engineering Research Council of Canada (NSERC; https://www.nserc-crsng.gc.ca/index_eng.asp) and the Canada Foundation for Innovation (CFI; https://www.innovation.ca/) (29438). The funders had no role in study design, data collection and analysis, decision to publish, or preparation of the manuscript.

**Competing interests:** The authors have declared that no competing interests exist.

(MWT) was developed as a direct behavioural measure of this type of spatial memory in rodents [2]. In this task, rats must learn and remember the location of a hidden platform submerged just below the surface in a circular pool. Successful navigation requires that rats use distal environmental cues (i.e., extra-pool cues) to find the location of the hidden platform making task performance dependent on allocentric spatial learning. When the HPC is damaged, rats take longer to locate the platform across acquisition trials compared to control rats and do not typically take a direct route to the platform location, suggesting an impairment in allocentric spatial learning [4–9]. Yet, rats with HPC lesions do show some improvement in locating the platform over trials in this task by resorting to a different navigation approach [5–7,10]. Indeed, analyses of swim strategies of rats with HPC lesions in the MWT show that they often adopt a stereotyped concentric swim pattern and monitor their distance from the pool wall to locate the platform rather than using distal cues to triangulate the location [4,7,8,11,12]. These rats learn to rely on an egocentric navigation strategy whereby locations are encoded relative to the animal's own position, a process that does not appear to require the HPC [4,7,13–15].

Although HPC lesions cause rats to adopt a different spatial learning and navigation strategy, it is unclear whether this effect is related to lesion size. Some studies have found a near-linear relationship between the amount of HPC damage and MWT impairment [5,16], suggesting that larger lesions may exacerbate allocentric deficits and promote reliance on egocentric strategies. These studies, however, did not examine strategy use, therefore it remains unclear whether lesion size predicts the transition from allocentric to egocentric navigation. Hence, we examined how HPC lesion size influences the use of allocentric and egocentric strategies in the MWT. Specifically, we aimed to determine whether the MWT performance of rats with HPC lesions improves because of the development of an egocentric swim strategy and whether this development is correlated with lesion extent. To address this, we performed swim path analyses to examine search strategies in the MWT in rats with and without HPC lesions and examined whether navigation strategy varied with lesion size. We found that a subgroup of HPC lesion rats showed consistent allocentric navigation and that the number of rats in this subgroup was comparable to the number that reliably adopted an egocentric strategy. In addition, this finding was not explained by lesion size. These findings suggest that allocentric navigation can be supported by other brain regions in the absence of the HPC, adding nuance to current understanding of the neural mechanisms underlying spatial navigation.

## Methods

### Subjects

The current study reports on archival data obtained from separate experiments conducted in our laboratory (between 2019–2022, each 1–5 months in duration) that focused on the role of the HPC in context learning and then ended with a spatial memory test in the MWT [17–19, manuscript in preparation]. All procedures for these experiments were approved by Trent University's Animal Care Committee, and followed the guidelines set by the Canadian Council of Animal Care (Protocol Number:

25361). Analgesics and anesthetics were used to minimize suffering. Humane endpoint criteria were defined as an animal experiencing significant weight loss (>15%), decrease in grooming behaviours, decrease in mobility and/or signs of discomfort persisting despite treatment efforts prescribed by the veterinarian. Across all experiments, humane endpoint criteria were respected and below the 5% rate approved in the protocol.

The study involved 68 male Long-Evans rats (Charles River, Quebec) approximately 3 months old at the start of behavioural training. The rats were pair housed in well-ventilated cages placed on a standard 12-hour light-dark cycle (lights turn on at 7:00 am), monitored daily, and were provided with 25-30g of rat chow daily, with water available ad libitum. 53 rats had HPC lesions (HPC group) whereas 15 were sham-operated control rats (SHAM group). The SHAM group comprised a subset of rats randomly and proportionally selected from all control groups in the experiments. Animals were stratified into low-, mid-, and high-performance tiers based on the distribution of MWT performance (% time spent in the target quadrant) across all SHAM rats from all experiments. Within each experimental group, 1–3 SHAM rats were randomly selected, with stratified sampling used to reflect the overall performance distribution of the SHAM rats while maintaining proportional representation across experimental cohorts. Prior to surgery, all rats were subject to contextual fear conditioning acquisition. Retention was tested following recovery, before the MWT was conducted.

## Apparatus

The MWT was conducted as previously described [20]. Briefly, testing occurred in a stainless-steel circular pool (150 cm in diameter, 92 cm in depth) filled with water maintained at 22 °C +/- 2 °C and rendered opaque using non-toxic white paint. A clear plexiglass platform (10 cm in diameter) was submerged 2 cm below the water's surface. The room contained multiple distal, extra-pool cues that were kept in constant configuration across all experiments. Behavioural data were recorded and analyzed using ANY-maze software (Stoelting, Wood Dale, IL).

## Surgery

All rats underwent surgery prior to MWT testing. The rats were anaesthetized with isoflurane (Abbott Laboratories, Chicago, IL) delivered in oxygen at 0.8–1.0 L/min (Benson Medical Industries, Markham, Ontario) for the duration of surgery, and received an analgesic (Metacam, 0.02 ml; 5 mg/ml, s.c.; Boehringer-Ingelheim, Rhineland-Palatinate, Germany). Once secured in a stereotaxic frame (Stoelting, Wood Dale, IL), a midline scalp incision was made and retracted to expose the skull and bregma. For the SHAM surgeries, the incision was sutured at this point, and the rat was placed in a recovery cage until ambulatory. For the HPC lesions, six small bilateral burr holes were drilled, and each site received infusion of a cocktail containing the excitotoxin N-methyl-D-aspartic acid (NMDA; 7.5 µg/µl, Sigma Chemical, St. Louis, MO) and the sodium channel blocker tetrodotoxin (TTX; 1ng/µl, in 0.9%, Sigma Chemical, St. Louis, MO) dissolved in 0.9% physiological saline. Surgical coordinates and infusion volumes followed those previously described [21,22] (see supporting information S1 Table for details).

Following infusion, the incision was sutured and the rat was placed in a heated recovery cage until ambulatory before being returned to the colony room. All rats were given an oral post-operative analgesic (Metacam, Oral Suspension 0.1 ml; 1.5 mg/ml, p.o.; Boehringer-Ingelheim) once daily for five days following surgery to minimize discomfort.

## Behavioural procedures

The MWT was conducted 2, 3, 4, or 20 weeks after surgery. The rats completed a single session of ten swim trials. The first nine trials were standard escape (acquisition) trials, with the platform always in the same location. The tenth trial was a probe test, in which the platform was removed to assess whether the rats developed a spatial bias for the former platform location. The pool was divided into four quadrants based on cardinal directions and the platform was always located in the middle of the northeast (NE) quadrant. At the start of each trial, rats were placed in the pool facing the wall of one of the non-platform quadrants with starting positions varied in a semi-random sequence. Each rat was given 60 s to locate

 

the platform and, upon reaching it, allowed to remain on the platform for 5 s before being removed from the pool. Rats that failed to locate the platform within 60 s were gently guided to it by the experimenter and allowed to remain on the platform for 5 s. During the probe test, the rats were allowed to swim for 30 s.

During the acquisition trials, path length (distance swam to the platform, in metres) was used as the primary measure because it is analogous to latency to reach the platform while also accounting for swim speed. Finally, we also considered additional metrics from the probe trial. This included island crossings (number of passes over the former platform location) and percent time in the target quadrant (quadrant that formerly housed the platform).

**Swim path classification**

A researcher blind to each rat's lesion status (SHAM vs HPC) classified each swim trial as allocentric, egocentric, or random by examining representative track plots generated by the recording software (see Fig 1 for examples). These categories were operationally defined based on behaviours described by Dalm et al., [23], however, the nomenclature used for categories was adjusted for the current study; swim paths consistent with the "Spatial Persistent" category are referred to as allocentric in the current study, those consistent with "Non-Spatial Concentric" are referred to as egocentric, and "Non-Spatial Random" as random. Specifically, short swim paths aimed directly to the platform location with little to no searching were classified as allocentric. Concentric swim paths leading to the platform location were classified as egocentric, whereas disorganized swim paths were classified as random.

Two approaches were used to measure the reliability of the swim path classification.

(1) A researcher blinded to the initial classification, trial number, and group (SHAM vs HPC); re-classified all swim trials using the operational definitions described above, with classification performed on full swim paths.

(2) RODA, a semi-supervised machine learning software developed by Gehring et al., [24] and Vouros et al., [25] was used to classify individual swim path segments within each trial as either allocentric, egocentric, or random.

For the machine learning classification, each swim path was divided into segments of a specified length. Previous work by Gehring et al., [24] and Vouros et al., [25] demonstrated that RODA performs well using segments of 2–3 m in length and 70–90% overlaps between segments. In the present study, the segments used were 2 m in length with 70% overlap. As recommended by Gehring et al. [24], approximately 15% of the total number of segments were initially manually classified to train the algorithm. Once trained by the labelled segments, RODA then performed the classification on the full sample.

Because RODA classifies multiple segments within each trial, the data needed further processing to allow comparison with the human classification method in which each trial was assigned by the researcher to a single category. To accomplish this, we calculated the proportion of segments within each trial assigned to each classification. A full swim trial was

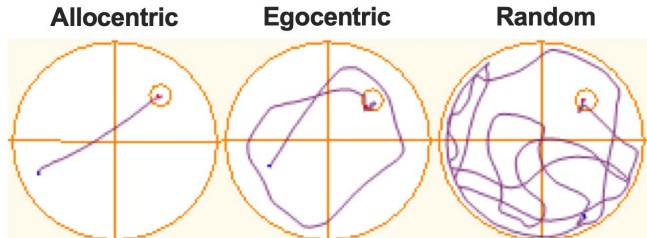

**Fig 1. Representative track plots depicting examples of allocentric, egocentric, and random swim paths respectively.** Track plots were generated using ANY-maze software. The orange lines represent the pool's perimeter, the platform location (smaller circle in the top-right quadrant), and the quadrant boundaries. The purple line represents the rat's swim path, beginning at the blue dot and ending at the red dot.

then labelled as allocentric or egocentric if more than 50% of its segments belonged to that category. For example, if four of five segments (80%) were classified as egocentric, then the whole trial was considered egocentric. Trials that did not meet the threshold for allocentric or egocentric were classified as random, as the primary objective of the current study was to identify consistent biases toward a given strategy (or lack thereof).

### HPC subgroups

After confirming the reliability of the researcher's initial classifications, HPC rats were divided into subgroups based on trial classification outcomes. Rats showing consistent allocentric strategy use were assigned to the allocentric subgroup (A-HPC), and those demonstrating consistent egocentric strategy use to the egocentric subgroup (E-HPC). Strategy use was considered consistent when the same strategy was employed in at least 75% of the final four acquisition trials. Rats that did not meet this criterion and exhibited random or inconsistent strategy use were placed in the random subgroup (R-HPC).

### Histology

Following the completion of behavioural testing, the rats were anesthetized with an intraperitoneal injection of sodium pentobarbital (0.3 mL; 340 mg/ml) and perfused intracardially with 200 mL of phosphate-buffered saline followed by 200 mL of 4% paraformaldehyde. The brains were removed and stored in 4% paraformaldehyde for 24 hours before being transferred to 0.1% sodium azide/30% sucrose solution to cryoprotect the tissue. The brains remained in the latter solution for a minimum of 48 hours until sectioned. The brains were then sectioned at a thickness of 40 μm using a cryostat (Slee, Mainz, Germany). Every twelfth section (sectioning sampling fraction of 1/12th) extending through the HPC was mounted onto Superfrost Plus glass microscope slides (Fisher Scientific, Hampton, NH), stained with Cresyl Violet, and cover slipped. Digital images of each section were then taken at a 2X magnification using a light microscope (Nikon H600L), camera (DS Qi1Mc), and Nikon Element software (Nikon Instruments Inc., Melville, NY), to enable quantification of the lesions.

The HPC lesion extent in each rat was estimated according to the Cavalieri and point-counting principles [26]. Using Fiji (ImageJ) software (https://imagej.net/Fiji), a sampling grid with an area per point of 0.05 mm² was randomly superimposed on each digitized section. The HPC was defined as spanning from −1.72 mm to −6.84 mm relative to Bregma [27]. Grid points that intersected spared tissue in HPC cell fields (CA1–3, hilus, and dentate gyrus; 10–12 sections per brain) were counted for each section. The average number of points across all SHAM brains was calculated. The total number of points counted for each lesion brain was divided by the SHAM average and multiplied by 100 to provide an estimated percent of HPC tissue spared. The compliment of the percent spared was then used to estimate lesion size.

### Statistical analysis

Data visualizations were created in GraphPad Prism (version 10.6.1), which was also used to conduct mixed analyses of variance (ANOVAs). All other analyses were conducted in R (version 4.5.1) using RStudio (version 2025.09.1). For non-parametric between-group comparisons, Kruskal-Wallis tests were used instead of ANOVAs when assumptions of normality or homogeneity of variance were violated, as determined by Shapiro-Wilk and Levene's tests. Wilcoxon signed-rank tests were used for non-parametric pairwise comparisons. Holm's adjusted $P$-values are reported for post hoc pairwise comparisons to control the family-wise error rate, unless the comparisons were planned, in which case the unadjusted $P$-values are reported. Chi-squared tests of independence were used to assess strategy use relative to chance on each trial, for each group. Pearsons's correlations were conducted to examine relationships between lesion size and behavioural performance. One-sample $t$-tests were used to compare percent time spent in the target quadrant on the probe trial against chance (25%). All statistical tests were two-tailed with $α = .05$.

## Results

### Histology

Infusions of the excitotoxin NMDA+TTX, in each lesion rat, produced extensive bilateral cell loss across all principal subfields (CA1–3 and dentate gyrus) of the HPC and across its septo-temporal axis (see Fig 2). The HPC lesion size ranged from 51.37% to 96.42% with an average of 84.15% ± 1.40%. The lesion distribution is illustrated in Fig 2. In animals with smaller lesions, sparing was primarily observed in the ventral hippocampus (approximate span: −4.08 mm to −6.84 mm relative to Bregma). No damage was observed in surrounding regions, including the thalamus or rhinal cortices.

### Inter-rater reliability of swim strategy classification

Two complementary approaches were employed to measure the reliability of swim strategy classification: inter-rater and RODA software comparisons. Overall, the two manual raters agreed on 87% of trials. RODA agreed with the initial classification on 77% of trials and with the second rater's classification on 74% of trials. Agreement was highest for trials initially classified as allocentric, with 86% agreement between raters and 99% agreement between the initial rater and the software. For trials initially classified as egocentric, agreement with the second manual rater was 81%, while agreement with RODA was 59%. For random trials, agreement with the second manual rater was 81%, and agreement with RODA was 59%. Further details are available in S2 Table. Given the high level of agreement between the primary experimenter, and both the second rater and the software, the initial manual ratings were deemed sufficiently reliable and used for all subsequent analyses.

### Acquisition trials

Of the 612 recorded acquisition trials, 14 were excluded due to inadequate tracking of swim paths, which prevented accurate strategy classification. Specifically, data were omitted from three rats in Trial 2 (HPC $n = 52$, SHAM $n = 13$), five in Trial 3 (HPC $n = 49$, SHAM $n = 14$), one in Trial 5 (HPC $n = 52$), two in Trial 6 (HPC $n = 52$, SHAM $n = 14$), and three in Trial 7 (HPC $n = 50$).

Fig 3 shows path lengths across the first nine swim trials for SHAM and HPC rats. A mixed design ANOVA was conducted with group (SHAM vs HPC) as the between-subjects factor and trial as the within-subjects factor. The interaction

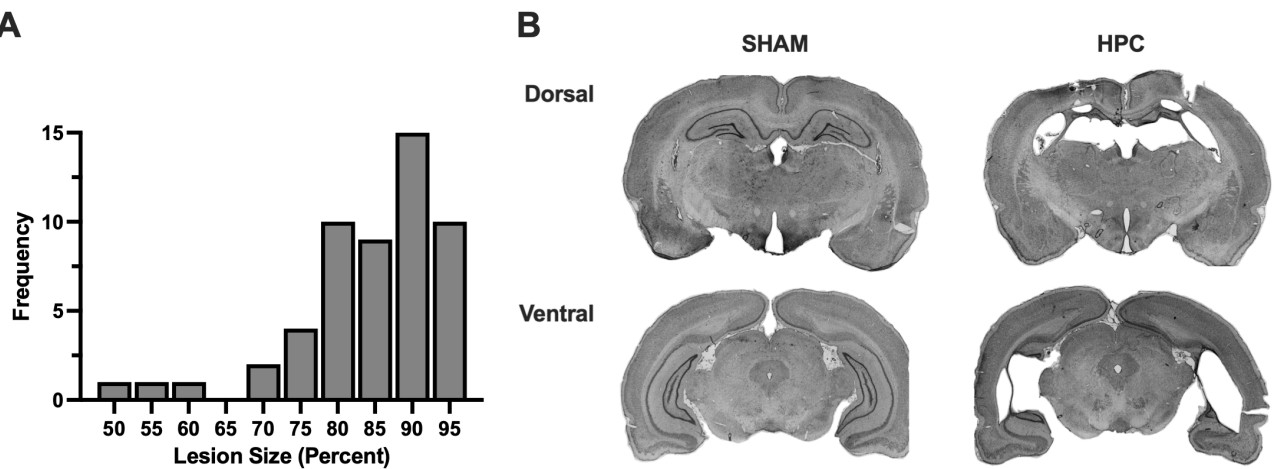

**Fig 2. HPC lesions. A)** Histogram depicting the lesion size distribution across all HPC rats. **B)** Representative photomicrographs of the intact HPC cell fields from a sham operated control (*left*) and similar sections from a lesion rat in which 88% of the HPC cell fields were damaged (*right*).

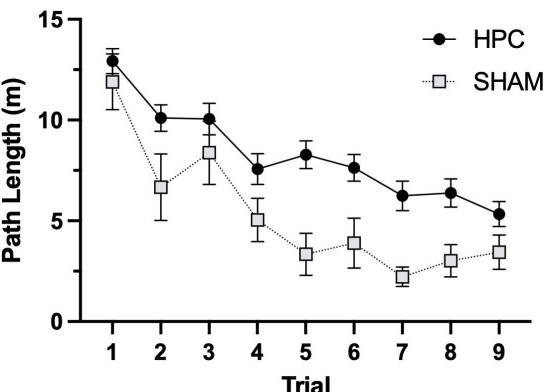

**Fig 3. MWT performance.** Mean (±SEM) distance travelled to the platform location (path length) in metres for each trial. Path lengths decreased significantly over trials ($P<.001$), suggesting that the rats learned to navigate to the platform to escape the water. The HPC group had significantly longer path lengths than SHAM ($P<.001$), suggesting that HPC lesions impaired their navigation strategy.

was not significant ($F_{8,515}=0.829$, $P=.577$); however, there was a significant main effect of trial ($F_{8,515}=15.23$, $P<.001$), indicating that path lengths decreased significantly over trials. The main effect of group was also significant ($F_{1,66}=25.23$, $P<.001$), indicating that the HPC group had significantly longer path lengths than the SHAM group.

Fig 4 depicts the relative frequency of navigation strategies used by each group. For each trial, a chi-square test of independence was used to determine whether strategy use differed from chance, i.e., whether a significant proportion of rats adopted a non-random search strategy. Trial 1 was excluded from our analysis as all rats displayed random swimming behaviours.

For the SHAM group, strategy use did not differ from chance on Trial 2 ($\chi^2_1=0.6$, $P=.439$), Trial 3 ($\chi^2_1=0.6$, $P=.439$), Trial 4 ($\chi^2_1=3.27$, $P=.071$), or Trial 6 ($\chi^2_1=3.27$, $P=.071$). However, they did significantly engage in a strategy on Trial 5 ($\chi^2_1=11.27$, $P<.001$), Trial 7 ($\chi^2_1=11.27$, $P<.001$), Trial 8 ($\chi^2_1=8.07$, $P=.005$), and Trial 9 ($\chi^2_1=5.4$, $P=.020$). A second chi-square test of independence was performed on the trials where strategy use was significant to test whether the SHAM rats preferentially used one specific strategy. The results showed a significant bias toward allocentric strategies on Trial 5

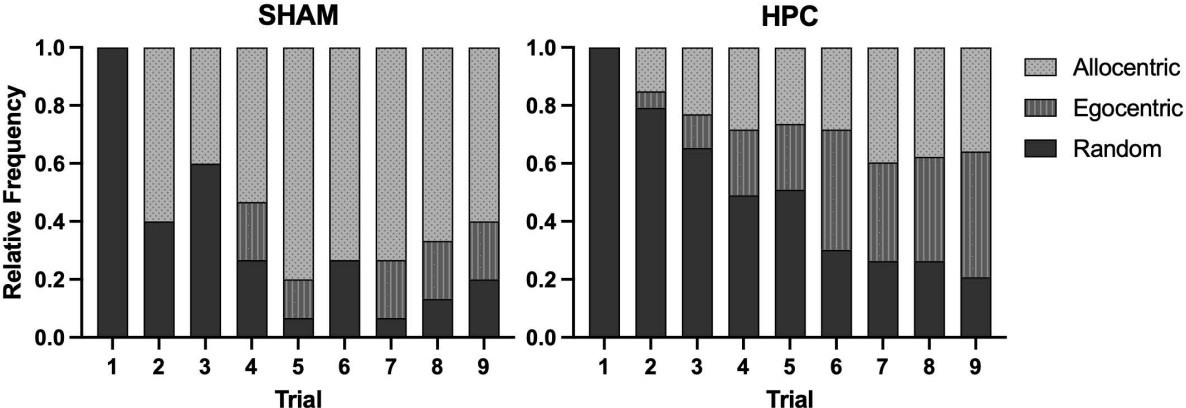

**Fig 4. Navigation strategy use.** Distribution of navigation strategies within each trial for SHAM (*left*) and HPC (*right*). The SHAM group demonstrated significant strategy use on Trials 5 ($P<.001$), 7 ($P<.001$), 8 ($P<.01$), and 9 ($P<.05$). A bias towards allocentric strategies was demonstrated on Trials 5 ($P<.001$), 7 ($P<.01$), and 8 ($P<.05$). The HPC group demonstrated significant strategy use on Trials 6 ($P<.01$), 7 ($P<.001$), 8 ($P<.01$), and 9 ($P<.001$). The HPC group did not demonstrate a bias towards allocentric or egocentric strategies ($Ps>.05$).

($\chi^2_2 = 14.8$, $P < .001$), Trial 7 ($\chi^2_2 = 11.2$, $P = .004$), and Trial 8 ($\chi^2_2 = 7.6$, $P = .022$), but not on Trial 9 ($\chi^2_2 = 4.8$, $P = .091$). Thus, SHAM rats begin to show a preference for allocentric search strategies by Trial 5 which largely persisted thereafter.

In contrast, a significant proportion of HPC rats engaged in random swim on Trial 2 ($\chi^2_1 = 18.13$, $P < .001$) and Trial 3 ($\chi^2_1 = 4.92$ $P < .001$). Strategy use did not differ from chance on Trial 4 ($\chi^2_1 = 0.17$, $P = .680$) or Trial 5 ($\chi^2_1 = 0.02$, $P = .891$). However, by Trial 6, they significantly engaged in strategy use (Trial 6, $\chi^2_1 = 8.32$, $P = .004$; Trial 7, $\chi^2_1 = 11.79$, $P < .001$; Trial 8, $\chi^2_1 = 11.79$, $P < .001$; Trial 9, $\chi^2_1 = 18.13$, $P < .001$). Despite this shift, strategy use among the HPC group did not bias towards the egocentric or allocentric strategy (Trial 6, $\chi^2_1 = 1.62$, $P = .444$; Trial 7, $\chi^2_1 = 1.40$, $P = .500$; Trial 8, $\chi^2_1 = 1.17$, $P = .557$; Trial 9, $\chi^2_1 = 4.23$, $P = .121$). Thus, the HPC group began showing non-random search by Trial 6, but unlike the SHAM group, without a clear delineation towards one strategy.

To determine whether individual HPC rats showed consistent use of a specific strategy, the HPC group was categorized into subgroups based on the results of swim path classification. Rats were assigned to the allocentric (A-HPC), egocentric (E-HPC), or random (R-HPC) subgroups depending on whether they demonstrated predominantly allocentric or egocentric strategies on at least 75% of the final four acquisition trials. The final four trials were selected for defining subgroup assignment because this was the stage at which a substantial proportion of HPC rats began to exhibit more consistent strategy use (see Fig 4). All remaining HPC rats were assigned to the R-HPC subgroup. Of the 53 HPC rats, 8 (15.09%) were assigned to A-HPC, 10 (18.87%) to E-HPC, and 35 (66.04%) to R-HPC. In the SHAM group, 9 rats (60%) used an allocentric strategy for the majority of the last four trials.

Fig 5A depicts the path lengths across each trial with the HPC group categorized into strategy-based subgroups. A mixed-design ANOVA was conducted with group (SHAM, A-HPC, E-HPC, R-HPC) as the between-subjects factor and trial as the within-subjects factor. The interaction was not significant ($F_{24,499} = 1.14$, $P = .292$), however, there was a significant main effect of group ($F_{3,64} = 13.95$, $P < .001$), and a significant main effect of trial ($F_{8,499} = 17.11$, $P < .001$). To evaluate group (SHAM, A-HPC, E-HPC, R-HPC) differences once initial learning had occurred, a one-way between-subjects ANOVA was performed using the mean path length across the final four trials (6–9). As depicted in Fig 5B, the effect of group was significant ($F_{3,64} = 13.28$, $P < .001$). Pairwise comparisons revealed that, when compared to the SHAM group, both the E-HPC subgroup ($P = .031$) and the R-HPC subgroup ($P < .001$) had significantly longer mean path lengths. In contrast, performance of the A-HPC did not significantly differ from that of the SHAM ($P = .855$).

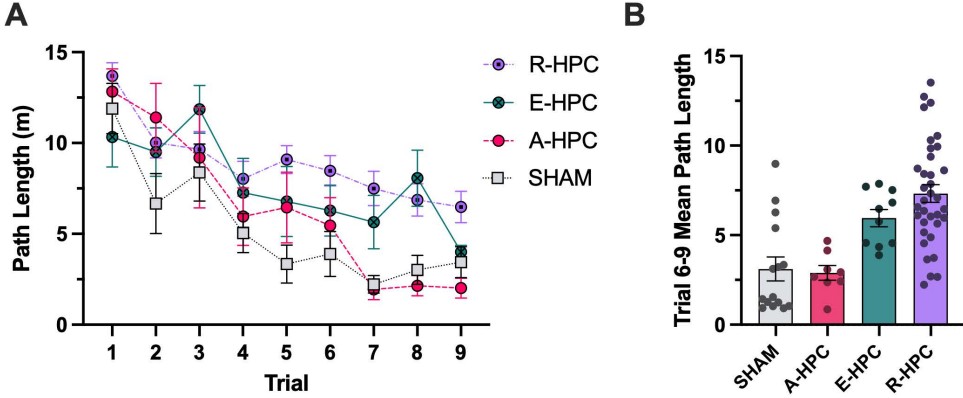

**Fig 5. MWT performance by subgroup. A)** Mean (±SEM) distance travelled to the platform location (path length) in metres for each trial. The HPC group was divided into 3 subgroups determined by consistent strategy use. Path lengths decreased significantly across trials ($P < .001$), suggesting that significant learning occurred. There was a significant main effect of group ($P < .001$), indicating that performance differed between groups. **B)** Mean (±SEM) path length for each subject's trial 6-9 average. Compared to SHAM, the E-HPC ($P < .05$) and R-HPC ($P < .001$) had significantly longer average path lengths. The A-HPC group did not differ from SHAM ($P > .05$).

A Pearson's correlation test was conducted between lesion size and the mean path lengths across Trials 6−9 to determine whether impairments resulted from larger lesions. Despite the HPC rats showing differing swim strategy use, lesion size was not significantly correlated with mean path length ($r_{51}$=0.037, $P$=.791). Additionally, there were no significant correlations between lesion size and mean path length within any of the HPC subgroups (A-HPC, $r_6$=−0.503, $P$=.204; E-HPC, $r_8$=−0.367, $P$=.297; R-HPC, $r_{33}$=−0.056, $P$=.751). As depicted in Fig 6, subgroups appeared similarly distributed across the lesion size distribution.

### Probe trial

To compare the distance swam to the platform (path length) on Trial 10 (once it had been removed) between subgroups, a Kruskal-Wallis test was conducted. As depicted in Fig 7A, the overall difference between groups was significant ($\chi^2_3$=8.86, $P$=.031). Pairwise comparisons revealed that none of the HPC subgroups differed significantly from SHAM (A-HPC $P$=1; E-HPC $P$=.403; R-HPC $P$=.063).

To determine whether groups persistently searched the location where the platform had been, a Kruskal-Wallis test was used to compare the number of times it was crossed. As depicted in Fig 7B, there was a significant effect of group ($\chi^2_3$=14.21, $P$=.003). Pairwise comparisons revealed a significant difference between the R-HPC and SHAM ($P$=.005), indicating that the R-HPC crossed the platform location significantly fewer times than SHAM. Neither the E-HPC nor the A-HPC differed significantly from SHAM (A-HPC $P$=.544; E-HPC $P$=.446).

The percent time spent in the target quadrant for each subgroup was compared to chance performance (25%) using one-sample $t$-tests to determine whether groups perseverated in this quadrant. Fig 7C shows that the SHAM group spent significantly more time in the target quadrant than would be expected by chance ($t_{14}$=3.21, $P$=.003), however none of the HPC subgroups performed significantly above chance (A-HPC $t_7$=0.953, $P$=.186; E-HPC $t_9$=−1.13, $P$=.857; R-HPC $t_{34}$=−1.51, $P$=.930).

### Discussion

The primary aim of this study was to determine whether HPC damage extent predicts navigation strategy use in the MWT. Contrary to expectations, neither strategy use nor task performance correlated with lesion size. Whereas SHAM rats predominantly adopted allocentric navigation, HPC rats showed mixed patterns: similar proportions used allocentric or egocentric strategies, and most (66%) exhibited no clear preference. Notably, approximately 15% of rats with extensive

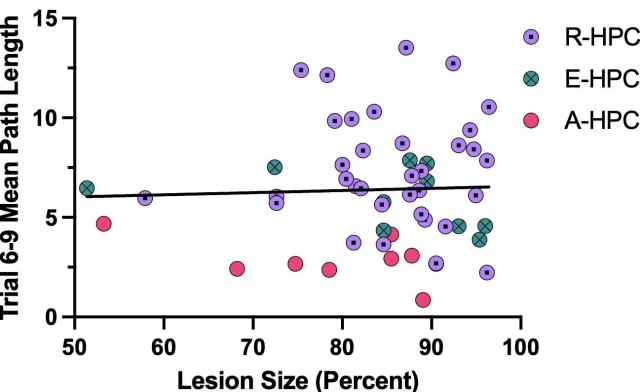

**Fig 6. Relationship between lesion size and task performance.** HPC lesion size compared to mean path length across Trials 6-9. The line of best fit is based on the HPC group overall. The correlation between lesion size and mean path length was not significant for the HPC group overall, nor within any of the HPC subgroups ($P$s > .05), indicating that lesion size does not predict performance.

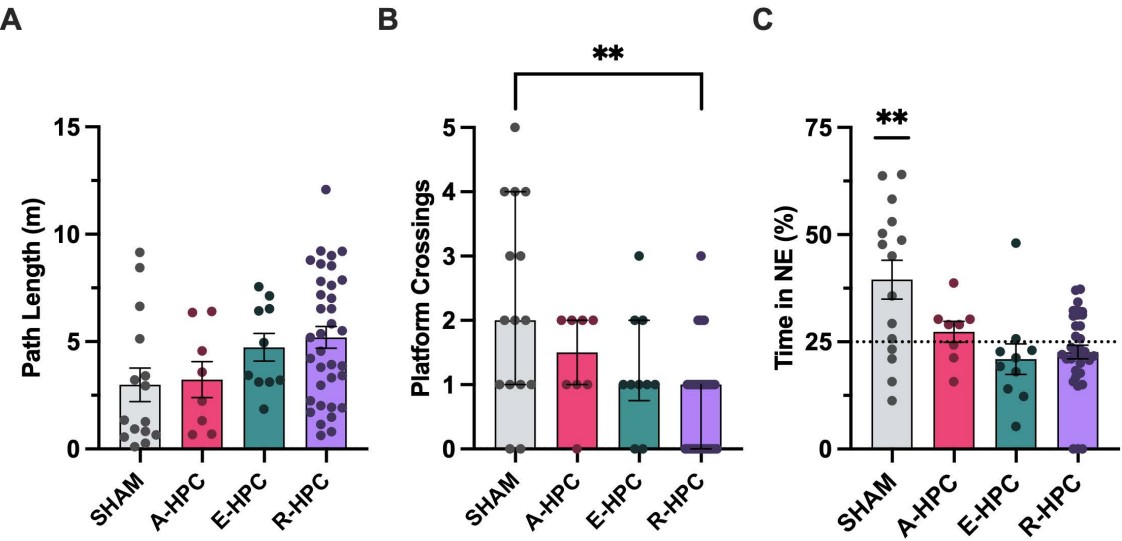

**Fig 7. Probe trial performance by subgroup.** Performance on the 30 second probe trial in which the platform was removed from the pool. **A)** Mean (±SEM) distance travelled to the platform location (path length) in metres. Total distance swam in metres is shown instead for subjects that did not cross the platform. Path length did not differ significantly between SHAM and any HPC subgroups (Ps > .05). **B)** Median (±IQR) number of times rats crossed the platform location. The R-HPC crossed the platform location significantly fewer times than SHAM (P < .01). The E-HPC and A-HPC subgroups did not differ significantly from SHAM (Ps > .05). **C)** Mean (±SEM) percent time spent in the NE (target) quadrant. The dotted line indicates chance performance (25%). The SHAM group spent significantly more time in the NE quadrant than would be expected by chance (P < .01). None of the HPC subgroups performed above chance (Ps > .05).

HPC damage (>85%) demonstrated intact allocentric learning, suggesting that other brain regions can support allocentric navigation in the absence of the HPC.

Consistent with prior literature, HPC damage impaired overall MWT performance, reinforcing the HPC's well-established role in spatial memory [2,3,7]. In addition to conventional metrics of MWT performance, we applied a swim path classification technique to examine the navigation strategy use. Inter-rater reliability was high between human raters and showed generally high agreement with the RODA classification software. Agreement was found to be highest for the allocentric trials and lowest for the egocentric trials. Importantly, our classification of swim paths revealed nuanced strategy use. SHAM rats developed a strong allocentric bias by Trial 5, aligning with previous findings that allocentric representations emerge early in training [28]. In contrast, HPC rats did not show a systematic shift toward egocentric navigation, as reported in earlier studies [4,7,13–15]. Instead, these rats adopted strategies early, and the proportions using allocentric and egocentric strategies were comparable. When applying a less conservative criterion (>60% consistency across the final five trials), 25% of HPC rats were classified as allocentric, and 23% as egocentric, reinforcing that reliable strategy use was emerging in almost half of the lesion rats. Moreover, HPC rats that consistently used allocentric navigation performed similarly to SHAM rats, whereas egocentric and random subgroups did not.

We anticipated that lesion size would influence navigation strategy, with smaller lesions preserving allocentric navigation and larger lesions favouring egocentric strategies. Contrary to this hypothesis, lesion size did not predict path length or strategy use, diverging from prior studies that reported a strong correlation between lesion extent and MWT impairment [5,16,29]. For example, Broadbent et al. [5] observed pronounced deficits when HPC damage exceeded 50%, yet in our sample, the smallest lesion was 51%, potentially limiting detection of this relationship. A correlation between lesion size and performance was anticipated despite this constraint, however, rats with near-complete lesions (>85%) not only adopted consistent allocentric strategies but also performed comparably to SHAM controls. This pattern suggests that allocentric learning can persist independently of HPC integrity through functional recruitment across other neural circuits.

Given the absence of allocentric HPC rats with lesions larger than 90%, it is possible that larger lesions would eliminate allocentric strategy use. However, it is unlikely that performance could have been supported by the small islands of spared tissue, regardless of location [5,29,30]. The remnants of HPC tissue were limited to isolated patches, making it improbable that these fragments could meaningfully contribute to performance. Lesion extent was not conducive to subfield-specific quantification, though evidence from Hunsaker & Kesner [31] suggests that CA3-specific lesions would produce the most substantial deficits, followed by CA1 and the dentate gyrus. Lesions restricted to the CA2 subfield would not likely produce substantial deficits [32].

Although allocentric navigation has been observed in HPC rats, prior studies suggest this typically requires extensive training or environmental familiarity [9,33]. Our results indicate that HPC damage does not preclude early engagement in allocentric navigation. However, probe trial performance revealed weak perseverance in the target quadrant, suggesting poor memory for the platform location. Additional training might have strengthened this memory. In addition, allocentric navigation does not necessarily imply a full allocentric representation. HPC rats can learn to approach a single cue [11,34–36], but such stimulus-response learning cannot fully explain our findings. Prior studies show that HPC damage impairs performance even when a salient beacon marks the platform [37–39], indicating that allocentric learning persists but strategic integration is compromised.

Supporting this interpretation, studies of cued navigation reveal that when a proximal cue is removed and only distal cues remains, rats with HPC damage can still navigate to the same location [35,36]. These findings suggest that allocentric spatial learning can occur in the absence of the HPC under less computationally demanding conditions. In contrast, optimal performance in the hidden-platform version of the MWT requires integrating information from multiple sources to coordinate navigation behaviour [9,28,34–36,40]. Thus, an allocentric spatial representation is only one component within the complex computational processes that support task performance. Our results highlight that allocentric representations can be formed and, in some cases, integrated for navigation despite extensive HPC damage.

Allocentric navigation is increasingly understood as a product of distributed neural systems rather than the HPC alone. The retrosplenial cortex appears central to this process transforming egocentric reference frames into allocentric ones, forming landmark-based spatial representations, and integrating inputs from visual and memory networks [41–43]. These functions position the retrosplenial cortex as a critical hub for coordinating spatial information across multiple modalities. In contrast, egocentric navigation is thought to rely on regions such as the medial entorhinal cortex, postrhinal cortex, and dorsolateral striatum, which encode spatial boundaries and positional relationships [14,41]. Our findings suggest that allocentric navigation in HPC rats may emerge through these alternative circuits, likely using simplified spatial reference frames rather than the fully integrated representations typically associated with HPC processing [2,3,7,12,14]. This perspective highlights the flexibility of spatial networks and underscores the importance of examining non-HPC contributions to navigation.

## Conclusion

This study demonstrated that MWT performance was not related to HPC damage extent and that a subset of rats consistently used allocentric strategies despite extensive HPC damage. These results challenge the long-standing view that allocentric learning is uniquely dependent on the HPC and highlight the distributed nature of spatial memory systems. Such findings have implications for how MWT performance is measured and interpreted and, more broadly, for theoretical models of memory organization.

## Supporting information

**S1 Fig. Performance between different surgery–behavioural testing intervals.** Mean (±SEM) path length for each subject's trial 6–9 average. A one-way ANOVA revealed no significant differences between the different surgery–behavioural testing intervals ($F_{3,49}=2$, $P=.126$).
(TIFF)

**S2 Fig. Subgroup distribution within different surgery–behavioural testing intervals.** Percentage of rats assigned to allocentric, egocentric, and random subgroups for each interval. Chi-square tests for independence determined that the distribution of subgroups did not significantly differ between intervals (A-HPC $\chi^2_3 = 2$, $P = .572$; E-HPC $\chi^2_3 = 3.6$, $P = .308$; R-HPC $\chi^2_3 = 1.69$, $P = .640$).
(TIFF)

**S1 Table. Stereotaxic injection coordinates relative to Bregma (mm) and volume of NMDA+TTX infused at each site.**
(DOCX)

**S2 Table. Rater agreement on swim path classification.**
(DOCX)

## Acknowledgments

We thank Kassidy Roberts for her contributions to conducting the behavioural experiments. Artificial intelligence tools (Chat GPT Version 1.2025.329, Microsoft Copilot) were used for editing purposes. All hypotheses, interpretations, results, conclusions, limitations and implications of the study; however, reflect the author's own ideas.

## Author contributions

**Conceptualization:** Jordan A. Webb, Neil M. Fournier, Michael G. Reynolds, Liana E. Brown, Hugo Lehmann.

**Formal analysis:** Jordan A. Webb.

**Funding acquisition:** Hugo Lehmann.

**Methodology:** Jordan A. Webb, Hugo Lehmann.

**Project administration:** Jordan A. Webb, Hugo Lehmann.

**Resources:** Hugo Lehmann.

**Software:** Jordan A. Webb, Sebastien Paquette.

**Supervision:** Hugo Lehmann.

**Visualization:** Jordan A. Webb.

**Writing – original draft:** Jordan A. Webb.

**Writing – review & editing:** Jordan A. Webb, Sebastien Paquette, Neil M. Fournier, Michael G. Reynolds, Liana E. Brown.

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
