## [Decision Letter · Decision Letter 0]

1 Feb 2026

Dear Dr. Webb,

Please see comments from the Editor and from the Reviewers below.

We look forward to receiving your revised manuscript.

Kind regards,

Miriam A. Hickey, PhD

Academic Editor

PLOS One

**Journal Requirements:**

2. We note that your Data Availability Statement is currently as follows:

“All relevant data are within the manuscript and its Supporting Information files.”

4. Please upload a new copy of Figure 1  as the detail is not clear. Please follow the link for more information:  https://journals.plos.org/plosone/s/figures

**Additional Editor Comments:**

Editor Comments:

Please now address all comments from the Reviewers and also comments from the Editor.

Comments from Editor

Methods

Subjects

Please provide more information on these selections, to demonstrate adherence to ARRIVE recommendations.

Surgery

Although already reported, please provide information on approximate bregma levels where lesion occurred.

Please also provide the time between surgery and behavioural task.

Behavioural Procedures

Please justify the reason for not including Gallagher’s proximity, e.g., https://doi.org/10.3389/neuro.07.004.2009

Results

Probe Trial

Please indicate this difference on the graph, e.g., Fig 7b, as asterisk or other symbol.

Reviewers' comments:

Reviewer's Responses to Questions

**Comments to the Author**

1. Is the manuscript technically sound, and do the data support the conclusions?

Reviewer #1: Partly

Reviewer #2: Partly

2. Has the statistical analysis been performed appropriately and rigorously?

Reviewer #1: Yes

Reviewer #2: Yes

3. Have the authors made all data underlying the findings in their manuscript fully available?

Reviewer #1: Yes

Reviewer #2: Yes

4. Is the manuscript presented in an intelligible fashion and written in standard English?

Reviewer #1: Yes

Reviewer #2: Yes

Reviewer #1: If we interpret correctly, the authors aim for the over-arching, long-standing, and still unanswered question: If a linearly increasing number of hippocampal neurons is ablated or silenced, when does hippocampus-dependent allocentric navigation “collapse” into other, non-allocentric strategies?

To date, no study has quantitatively mapped a graded, linear loss of hippocampal neurons to an exact behavioral collapse point in allocentric navigation (i.e., a specific threshold at which hippocampal strategies fail and animals switch to non-allocentric strategies).

In this context, although the authors do not answer the above question, we find the study interesting and believe it merits publication.

An additional strength of the study is its use of existing datasets, which avoids expending additional resources and time.

We have, however, the following three main points that we recommend the authors address prior to publication.

Point 1: Temporal dynamics in strategy classification

Regarding machine-learning analysis with RODA, the authors correctly note that animals change strategies throughout individual learning sessions and during the probe trial. Collapsing behavioral dynamics into a fixed classification of individual whole trials risks obscuring lesion-related behavioral adaptations that are among the most informative aspects of the dataset.

We suggest providing summary statistics on temporal, path-related strategy dynamics across individual animals. Specifically, RODA segment-wise allocentric versus non-allocentric scores could be analyzed over trial time within individual trials, allowing readers to appreciate segment-wise allocentric–egocentric transitions. Under this approach, the “random” category would be unnecessary and could be reinterpreted as a meaningful, highly informative group. These analyses could be presented alongside the existing results to provide a richer picture of intra-trial behavioral dynamics.

Point 2: Trial-segment analysis

It is possible that stereotypical temporal distributions of strategies exist within trials, including the patterns of transitions between them. Establishing a new variable, such as “trial segment,” would enable RODA classifications to be evaluated across consecutive ordered trials (1–9). Subsequent statistical analyses could then reveal interactions among factors such as SHAM versus different lesion degrees, and how these factors influence intra-trial strategy dynamics across trials.

Point 3: Interpretation of size-strategy correlations

The current interpretation that “the absence of a size-strategy correlation is striking” is overly simplistic. Each hippocampal lesion constitutes a temporal-anatomical path with often unpredictable behavioral outcomes. For instance, an animal in which a given CA1 subpopulation “dies first” as a result of the individual procedure, cannot easily be compared with an animal in which similar changes occur first in CA3 or CA2. We recommend that the authors elaborate on this aspect in the discussion, highlighting the inherent variability and path-dependence of hippocampal lesions.

Reviewer #2: The time line of surgery and behavior is unclear from the manuscript. The surgery methods describe administering Metacam post-operatively once daily for 5 days, but it is unclear whether this is the total amount of time between surgery and behavior testing. A complete description and figure showing the time line of all manipulations is necessary. The work is said to be based on archival data of previously conducted experiments, but the experiments cited (citations 17-19) appear to be conference abstracts and not peer reviewed papers fully describing the experiments.

Figure 1 shows example plots of allocentric, egocentric and random swim paths, but the reviewer has questions about classification of the random swim paths. Based on the description on line 186-187, it seems that any trials that do not meet the criteria of allocentric or egocentric are automatically classified as random. The classification criteria implies that a swim trial must be labeled with >50% of segments as allocentric to be considered allocentric or >50% as egocentric to be considered egocentric. This criteria seems to imply that a trial that consisted of 40% allocentric and 40% egocentric segments would be considered random. While this may be an edge case, could it contribute to the discrepancy between RODA and the assessments by the first manual rater for egocentric and random trials (59% agreement in each case, versus 99% agreement on allocentric trials)? Should there be a revised criteria to handle for edge cases where the majority of the trial is made up of a combination of allocentric and egocentric rather than random motion? At the very least there should be some discussion of why this is not necessary if not. Additionally, the agreement between the second manual rater and RODA does not appear to be reported. It would be helpful to have a graph of the trial assessment overlap between both of the manual raters as well as RODA for each of the three categories.

Although figure 6 shows a lack of correlation between lesion size and mean path length, it appears that A-HPC are absent above about 90% lesion size. This appears to warrant some discussion on whether fully complete ablation of the hippocampus would prohibit the allocentric strategies used within the A-HPC subgroup.

It is also noteworthy that while the A-HPC subgroup did not exhibit a significantly longer path length on trials 6-9 compared to sham in figure 5B, the time spent in the NE quadrant during the probe trial in figure 7C is still no better than chance on average for any of the subgroups, including A-HPC. The conclusion that "MWT performance was not related to HPC damage extent" is overly strong, particularly without a more in depth analysis of how lesion size may have correlated to the stratification of search strategy subgroups. That a subset of rats consistently used allocentric strategies is compelling, but more specific analysis should be presented to show whether or not mice with >90% hippocampal lesion size are under-represented within this subgroup.

**Do you want your identity to be public for this peer review?** For information about this choice, including consent withdrawal, please see our Privacy Policy

Reviewer #1: **Yes:** Miguel Remondes

Reviewer #2: No

---

## [Author Response · Author response to Decision Letter 1]

16 Feb 2026

Response to Editor:

Subjects: Please provide more information on these selections, to demonstrate adherence to ARRIVE recommendations.

Further elaboration has been provided (lines 106-112) to demonstrate adherence to ARRIVE recommendations. Here is the specific statement:

“The SHAM group comprised a subset of rats randomly and proportionally selected from all control groups in the experiments. Animals were stratified into low-, mid-, and high-performance tiers based on the distribution of MWT performance (% time spent in the target quadrant) across all SHAM rats from all experiments. Within each experimental group, 1-3 SHAM rats were randomly selected, with stratified sampling used to reflect the overall performance distribution of the SHAM rats while maintaining proportional representation across experimental cohorts. Prior to surgery, all rats were subject to contextual fear conditioning acquisition. Retention was tested following recovery, before the MWT was conducted.”

Surgery: Although already reported, please provide information on approximate bregma levels where lesion occurred.

Surgical coordinates and infusion volumes are now provided as supporting information (Table S1).

Surgery: Please also provide the time between surgery and behavioural task.

The time between surgery and the behavioural task has been specified (line 141), which is as follows:

“The MWT was conducted 2, 3, 4, or 20 weeks after surgery.”

The data are depicted in the graph below, and the effects were not significant. We’ve included this as well as a graph depicting subgroup distribution per interval as supporting information.

Behavioural Procedures: Please justify the reason for not including Gallagher’s proximity.

Gallagher’s proximity (cumulative distance to the platform) was initially analyzed, however, the outcome was similar to that of path length (graphs below). Given the redundancy we did not believe that it was necessary to include as it does not add to the overall results.

Results (Probe Trial): Please indicate this difference on the graph, e.g., Fig 7b, as an asterisk or other symbol.

The significant results are now indicated on the graphs in Fig 7.

Response to Reviewer #1:

Regarding machine-learning analysis with RODA, the authors correctly note that animals change strategies throughout individual learning sessions and during the probe trial. Collapsing behavioral dynamics into a fixed classification of individual whole trials risks obscuring lesion-related behavioral adaptations that are among the most informative aspects of the dataset. We suggest providing summary statistics on temporal, path-related strategy dynamics across individual animals. Specifically, RODA segment-wise allocentric versus non-allocentric scores could be analyzed over trial time within individual trials, allowing readers to appreciate segment-wise allocentric–egocentric transitions. Under this approach, the “random” category would be unnecessary and could be reinterpreted as a meaningful, highly informative group. These analyses could be presented alongside the existing results to provide a richer picture of intra-trial behavioral dynamics. It is possible that stereotypical temporal distributions of strategies exist within trials, including the patterns of transitions between them. Establishing a new variable, such as “trial segment,” would enable RODA classifications to be evaluated across consecutive ordered trials (1–9). Subsequent statistical analyses could then reveal interactions among factors such as SHAM versus different lesion degrees, and how these factors influence intra-trial strategy dynamics across trials.

We appreciate the suggestion regarding an intra-trial assessment of behavioural dynamics. This is indeed an interesting direction for future work, however, such an analysis is beyond the scope of the current manuscript. While stereotypical temporal distributions of strategies–and the transitions between them–may well exist within trials, our present focus is restricted to broader strategy classification at the trial level. Ultimately, we view that this type of intra-trial analysis would not provide additional insight into the stable allocentric versus egocentric strategy classification that are the focus of our current study.

That being stated, we clarified in the manuscript the objective of the whole trial classification aiming to detect reliable biases. The rationale for this has been added to the methods section (lines 194-196), and is as follows:

“Trials that did not meet the threshold for allocentric or egocentric were classified as random, as the primary objective of the current study was to identify consistent biases toward a given strategy (or lack thereof).”

The current interpretation that “the absence of a size-strategy correlation is striking” is overly simplistic.

We agree that describing the absence of a size–strategy correlation as merely “striking” was overly simplistic. In the revised manuscript, we now provide a more nuanced interpretation (lines 410-411):

“A correlation between lesion size and performance was anticipated despite this constraint, however, rats with near-complete lesions (>85%) not only adopted consistent allocentric strategies but also performed comparably to SHAM controls.”

Each hippocampal lesion constitutes a temporal-anatomical path with often unpredictable behavioral outcomes. For instance, an animal in which a given CA1 subpopulation “dies first” as a result of the individual procedure, cannot easily be compared with an animal in which similar changes occur first in CA3 or CA2. We recommend that the authors elaborate on this aspect in the discussion, highlighting the inherent variability and path-dependence of hippocampal lesions.

This is an interesting recommendation, in fact, and an analysis we initially considered. Unfortunately, the lesions did not target specific subfields differentially, and therefore limited our ability to conduct subfield-specific analyses.

To address some of the concerns raised here, however, we now consider the implications for subfield-specific lesions in the discussion section (lines 415-419):

“Lesion extent was not conducive to subfield-specific quantification, though evidence from Hunsaker & Kesner [31] suggests that CA3-specific lesions would produce the most substantial deficits, followed by CA1 and the dentate gyrus. Lesions restricted to the CA2 subfield would not likely produce substantial deficits [32].”

Response to Reviewer #2:

The time line of surgery and behavior is unclear from the manuscript. The surgery methods describe administering Metacam post-operatively once daily for 5 days, but it is unclear whether this is the total amount of time between surgery and behavior testing. A complete description and figure showing the time line of all manipulations is necessary.

As described above in response to the editor, the time between surgery and behavioural testing has been clarified in the manuscript (line 141) and added to the supporting information (Figs S1, S2)

“The MWT was conducted 2, 3, 4, or 20 weeks after surgery.”

The work is said to be based on archival data of previously conducted experiments, but the experiments cited (citations 17-19) appear to be conference abstracts and not peer reviewed papers fully describing the experiments.

We appreciate the attention to the nature of the cited archival data. To clarify, all datasets used in the present manuscript originate from experiments previously completed in our laboratory. These data remain part of our internal archival repository regardless of whether the original studies were disseminated as peer-reviewed publications or conference abstracts. Their archival status refers to their origin within our lab’s historical dataset–not their publication format.

In addition, for the purposes of the current manuscript, all relevant procedures, including animal housing, surgical methods, MWT testing, and histological processing are described in complete detail within the Methods section.

Figure 1 shows example plots of allocentric, egocentric and random swim paths, but the reviewer has questions about classification of the random swim paths. Based on the description on line 186-187, it seems that any trials that do not meet the criteria of allocentric or egocentric are automatically classified as random. The classification criteria implies that a swim trial must be labeled with >50% of segments as allocentric to be considered allocentric or >50% as egocentric to be considered egocentric. This criteria seems to imply that a trial that consisted of 40% allocentric and 40% egocentric segments would be considered random. While this may be an edge case, could it contribute to the discrepancy between RODA and the assessments by the first manual rater for egocentric and random trials (59% agreement in each case, versus 99% agreement on allocentric trials)? Should there be a revised criteria to handle for edge cases where the majority of the trial is made up of a combination of allocentric and egocentric rather than random motion? At the very least there should be some discussion of why this is not necessary if not.

Similar to what was stated in response to Reviewer # 1 (see Point 1 above), looking at intra-trial components for classification may not strengthen results regarding the development of reliable biases, as was a primary aim of our study.

Additionally, the agreement between the second manual rater and RODA does not appear to be reported. It would be helpful to have a graph of the trial assessment overlap between both of the manual raters as well as RODA for each of the three categories.

The overall agreement between the second manual rater and RODA was 74%, and was reported on lines 258 and 259, though we did not provide a breakdown by category as we did for the other comparisons.

We have now compiled all of this information into a table provided in supporting information (S2).

Although figure 6 shows a lack of correlation between lesion size and mean path length, it appears that A-HPC are absent above about 90% lesion size. This appears to warrant some discussion on whether fully complete ablation of the hippocampus would prohibit the allocentric strategies used within the A-HPC subgroup.

We appreciate this recommendation so we have added the following statement to the manuscript to address this (lines 415-419):

“Given the absence of allocentric HPC rats with lesions larger than 90%, it is possible that larger lesions would eliminate allocentric strategy use. However, it is unlikely that performance could have been supported by the small islands of spared tissue, regardless of location [5,30,29]. The remnants of HPC tissue were limited to isolated patches, making it improbable that these fragments could meaningfully contribute to performance.”

Additionally, when we used the less conservative criterion (>60% consistency across the final five trials, as described on lines 400-403), the A-HPC subgroup did include rats with lesions larger than 90%.

It is also noteworthy that while the A-HPC subgroup did not exhibit a significantly longer path length on trials 6-9 compared to sham in figure 5B, the time spent in the NE quadrant during the probe trial in figure 7C is still no better than chance on average for any of the subgroups, including A-HPC. The conclusion that "MWT performance was not related to HPC damage extent" is overly strong, particularly without a more in depth analysis of how lesion size may have correlated to the stratification of search strategy subgroups.

We appreciate the reviewer’s observation regarding the probe trial outcomes. As noted in the initial submission, we discussed that the probe trial data likely reflect weak or imprecise spatial representation, and we highlighted the limitations of these measures in interpreting group differences (lines 424-427). We have ensured this caveat remains clear in the revised manuscript.

Our interpretation placed somewhat greater emphasis on the final four acquisition trials because averaging across multiple trials reduces the influence of trial-to-trial variability and chance performance fluctuations, thereby offering a more stable estimate of learning. Nonetheless, we agree that the probe trial provides complementary information, and we now clarify this relationship more explicitly.

The statement referenced by the reviewer pertains specifically to the correlation between mean path length on trials 6-9 and lesion size, which was the primary research question. As this relationship was not statistically significant, we aimed to reflect those findings accurately.

That a subset of rats consistently used allocentric strategies is compelling, but more specific analysis should be presented to show whether or not mice with >90% hippocampal lesion size are under-represented within this subgroup.

The concerns about the rats with lesions larger than 90% have been addressed in the manuscript (see Point 5 above).

---

## [Decision Letter · Decision Letter 1]

24 Feb 2026

Evidence of allocentric spatial learning in male rats with large lesions of the hippocampus

PONE-D-25-65950R1

Dear Dr. Webb,

We’re pleased to inform you that your manuscript has been judged scientifically suitable for publication and will be formally accepted for publication once it meets all outstanding technical requirements.

Kind regards,

Miriam A. Hickey, PhD

Academic Editor

PLOS One

Additional Editor Comments:

All Reviewer comments have been addressed satisfactorily.

Reviewers' comments:

Reviewer's Responses to Questions

**Comments to the Author**

Reviewer #1: All comments have been addressed

Reviewer #2: All comments have been addressed

2. Is the manuscript technically sound, and do the data support the conclusions?

Reviewer #1: Yes

Reviewer #2: Yes

3. Has the statistical analysis been performed appropriately and rigorously?

Reviewer #1: Yes

Reviewer #2: Yes

4. Have the authors made all data underlying the findings in their manuscript fully available?

Reviewer #1: Yes

Reviewer #2: Yes

5. Is the manuscript presented in an intelligible fashion and written in standard English?

Reviewer #1: Yes

Reviewer #2: Yes

Reviewer #1: Though not completely, the authors have sufficiently addressed my comments.

The manuscript is ready to be published.

Reviewer #2: Thank you for thoroughly addressing the concerns raised in my critique. My only outstanding issue is that I feel the conclusion would be better supported by a stratification of hippocampal lesion size, but that is now clarified as a known limitation of this study.

**Do you want your identity to be public for this peer review?** For information about this choice, including consent withdrawal, please see our Privacy Policy

Reviewer #1: **Yes:** Miguel Remondes

Reviewer #2: No

---

## [Editor Report · Acceptance letter]

PONE-D-25-65950R1

PLOS One

Dear Dr. Webb,

I'm pleased to inform you that your manuscript has been deemed suitable for publication in PLOS One. Congratulations! Your manuscript is now being handed over to our production team.

Kind regards,

on behalf of

Dr. Miriam A. Hickey

Academic Editor

PLOS One